**Uncertainty in regional estimates of capacity for carbon capture and storage**
Mark Wilkinson and Debbie Polson
School of GeoSciences, Grant Institute, James Hutton Road, The King's Buildings, The University
of Edinburgh, EH9 3FE
Correspondance to: mark.wilkinson@ed.ac.uk
**Abstract.** Carbon capture and storage (CCS) is a potentially important technology for
the mitigation of industrial $CO_2$ emissions. However the majority of the subsurface
storage capacity is in saline aquifers, for which there is relatively little information.
Published estimates of the potential storage capacity of such formations, based on
limited data, often give no indication of the uncertainty, despite there being
substantial uncertainty associated with the data used to calculate such estimates.
Here, we test the hypothesis that the uncertainty in such estimates is a significant
proportion of the estimated storage capacity, and should hence be evaluated as a
part of any assessment. Using only publicly available data, a group of 13 experts
independently estimated the storage capacity of 7 regional saline aquifers. The
experts produced a wide range of estimates for each aquifer due to a combination of
using different published values for some variables and differences in their
judgements of the aquifer properties such as area and thickness. The range of
storage estimates produced by the experts shows that there is significant
uncertainty in such estimates, in particular the experts' range does not capture the
highest possible capacity estimates. This means that by not accounting for
uncertainty, such regional estimates may underestimate the true storage capacity.
The result is applicable to single values of storage capacity of regional potential, but
not to detailed studies of a single storage site.


**1. Introduction**
Geological storage of carbon dioxide ($CO_2$) has been proposed as a potential
technological solution to help reduce emissions of greenhouse gases, given the
continued use of fossil fuels to meet much of the world's energy requirements. In
carbon capture and storage (CCS), the $CO_2$ produced from industrial sources is
captured and transported to a geological storage site and injected deep into the
subsurface where it is stored indefinitely in the pore space of the rocks. Saline
aquifers, rock formations where the pore space is filled with brines too saline for
useful extraction, offer the largest storage capacity (Holloway, 1997). However,
unlike hydrocarbon reservoirs, such formations often have limited legacy data. In
order to identify potential storage sites that are worth the investment required for
detailed assessment, attempts have been made to characterise regional saline
aquifers using this legacy data on both a regional and national scale. e.g. the NatCarb
Atlas for the USA (https://edx.netl.doe.gov/geocube/#natcarbviewer) and the
CO2Stored database for the UK (http://www.co2stored.co.uk). However care must
to taken to account for the substantial uncertainty associated with such regional
assessments. The capacity of a geological formation to store $CO_2$ securely is a first-
order concern in any storage assessment, and the basic methodology is well
established (Bachu, 2000). Lack of capacity is one of the highest risks to carbon
capture and storage projects (Polson et al., 2012) and uncertainty affects the design
of transport and injection networks (Keating et al., 2011; Middleton et al., 2012A;
Sanchez Fernandez et al., 2016).

Previous work on the subject includes the influence of estimated storage capacity
due to uncertainty in thermophysical properties (pressure and temperature of the
reservoir; Calvo et al., 2019). A rigorous Monte-Carlo approach has been
demonstrated using the $CO_2$-PENS tool (Keating et al., 2011), and it has been shown
that uncertainty in reservoir parameters can impact reservoir cost and capacity
estimates by as much as an order of magnitude (Middleton et al., 2012B). The
approach allows for the integration of site-specific data over a large range of size-
scales (Middleton et al., 2012A). Integrated Monte Carlo simulations constructed
using regional data have been used to assess $CO_2$ injectivity; the area of review;
migration rate into confining rocks; and the probability of detecting the injected $CO_2$
plume in monitoring wells (Dai et al., 2014). The state of the art is possibly the
integrated assessment model developed by the US DOE-funded National Risk
Assessment Partnership (Pawar et al., 2016). This approach has not yet been
universally adopted and cannot be easily applied retrospectively to pre-existing
studies.

Many published regional studies of $CO_2$ storage capacity quote single values for the
capacity of individual formations, sometimes with ranges allowing for uncertainty in
a single parameter such as the proportion of porespace that can be utilised for
storage ('storage efficiency') e.g. Medina et al. (2011). The reporting of individual
studies varies, but some provide storage estimates to 6 significant figures, implying a
precision of greater than 0.001 %. However, this precision is clearly unachievable,
since the commonly used methodologies for capacity calculation of saline aquifers
(e.g. Goodman et al. 2011) requires inputs which are inherently variable over the
area of assessment, such as the thickness of the formation, net:gross ratio (the
proportion of usable reservoir within the overall unit thickness), and porosity. When
offshore locations are considered, data are usually available from only a small
number of borehole penetrations, often with a spacing between boreholes of several
kilometres. While there are published methods for dealing with such uncertainty
(Burruss et al., 2009; Keating et al., 2011; Smith et al., 2011; Pawar et al., 2016),
estimates of the variability of each input parameter must be made, and suitable
software employed for the calculation. Consequently the use of single–value storage
estimates is both quicker and cheaper than full probabilistic assessments.

Furthermore, capacity assessments will largely depend on expert interpretation of
geological data, and are therefore dependent on the prior knowledge and
experience of individual experts (see Curtis, 2012, for summary). Studies have shown
that geological experts are subject to a range of cognitive biases, as are all
individuals (Kahneman et al., 1982), that combined with differences in prior
experience can influence their interpretation of data leading to subjective results
(e.g. Phillips, 1999; Polson and Curtis, 2010; Bond et al., 2012). As a result, an
estimate of the uncertainty of single-value storage capacities is of practical use, not
least with assessments already published but lacking an assessment of uncertainty.
This is of particular practical importance where a storage estimate falls close to a
cut-off value, below which, for example, a potential storage unit may be rejected as
having too low a storage capacity to be economically viable. For example, a regional
screening study (Wilkinson et al., 2010) rejected all units below an arbitrary 50 Mt of
estimated $CO_2$ storage capacity. For an individual storage project the minimum
acceptable storage capacity value is likely to be determined by the volume of $CO_2$ to
be stored over the project lifetime.

Here, we test the hypothesis that the uncertainty in storage estimates is a significant
proportion of the estimated storage capacity, and should hence be evaluated as a
part of any assessment. For this study, an assessment of the precision of storage
capacity estimates was conducted as part of a study of an area of the UK territorial
waters, in the Inner Moray Firth area of the North Sea (Fig. 1). Subsurface geological
data were available from boreholes drilled by the petroleum industry, both as
individual well records released by the UK Government, and summarised in scientific
publications. The subsea strata are largely siliciclastics, of Devonian to Jurassic age.
They rest unconformably on strata that were affected by the lower Palaeozoic
Caledonian Orogeny (Andrews et al., 1990), which are here considered to be
basement (i.e. to have no storage potential). To the east of the area, there is a
variable-thickness cover of Cretaceous Chalk, a fine-grained pelagic limestone, here
not considered as a potential store as it lacks an obvious seal. Questions concerning
the presence of a suitable seal, trapping structures and potential leakage pathways
were addressed in the wider study but are not reported here.

**2. Materials and Methods**

A group of 13 graduate students who had been trained in the methodology of
storage capacity estimation and in at least basic geology relevant to $CO_2$ storage,
assessed the capacity of the potential saline aquifers in the area. All the students
were studying for a Masters of Science degree in Carbon Capture and Storage, and
can be considered to be 'expert' in the subject, though their prior backgrounds are
variable ranging from geosciences to engineering.  The experts had to identify the
potential reservoir formations (saline aquifers) within the area using the scientific
literature, then collect the input information required to perform the basic storage

capacity estimates (surface area, thickness, porosity, net:gross ratio). The product of these parameters is an estimate of the volume of porewater within the aquifer, which may be compressed or partly displaced allowing for the storage of $CO_2$.

$$M=AhNG\Phi\rho E \qquad (1)$$

where M is the mass of $CO_2$ that can be stored, A is the area that defines the region being assessed, h is the thickness of the saline aquifer, NG is the net:gross ratio, $\Phi$ is the porosity, $\rho$ is the density of $CO_2$ and E is a storage efficiency factor.

For surface area the experts were directed to maps within Cameron (1993) and Richards et al. (1993); each expert independently estimated the area. Uncertainty in this parameter is therefore due to the variable interpretation of the same data from expert to expert. For the other parameters, the experts were expected to locate suitable data, primarily using web-based search tools. The uncertainty in these parameters is therefore determined by the total number and range of published values; the ease with which experts could find relevant information; and the interpretation by the experts of the applicability and reliability of the data that they located.

For the purposes of this paper, the values for each variable provided by the experts were combined with constant values of $CO_2$ density (650 kg/m$^3$) and storage efficiency (the proportion of porespace that can be utilised for storage, here taken to be 0.02), and the total storage capacities were re-calculated for each expert using Equation 1. This approach was undertaken to remove non-geological effects from the results, such as variation in estimated $CO_2$ density due to the use of different equations of state or pressure / temperature conditions of burial, and also any calculation errors. These individual estimates are hereafter referred to as experts' estimates however they are not the estimates calculated by the individual experts, but the estimates re-calculated by the authors using the data collected by each expert. For each geological unit, the standard deviation of the storage estimates was

calculated across the set of individual storage volume estimates. All experts gave
express permission for their data to be used for this purpose.

In order to determine the full range of possible estimates from the expert-derived
values, storage estimates were calculated for all possible combinations of the
variables. The resulting distribution of the storage estimates, *P(M)*, gives an
indication of their uncertainty. However as this method does not take into account
the real uncertainty in each variable (which is unknown), *P(M)* is not the probability
distribution of the storage capacity.

**3. Results**

There are 7 geological units (which are either Formations or Members in formal
nomenclature; Cameron et al., 1993; Richards et al, 1993) that are potential storage
reservoirs in the area, henceforth called storage units. Figure 2 shows *P(M)* as a
cumulative density function for each formation and Table 1 shows the median and
range of the individual expert estimates and the 5[th], 95[th] and median of P(M). Both
show a wide range of possible estimates for the storage capacity. The range of P(M)
is typically between 2 and 6 times the median value, though in the case of the Orrin
Formation, the range is 13 times the median.

The median values of the expert estimates tend to be similar to the median of the
distribution (within 10 %, except the Hopeman Sandstone which is within 20 %). The
individual expert estimates tend to cover the range from the 5[th] to 95[th] percentiles of
*P(M)*, though in 3 formations the minimum expert estimate exceeds the 5[th]
percentile of *P(M)* and in the case of the Hopeman Sandstone Formation, the lowest
expert estimate as at around the 15[th] percentile. For 2 formations, the maximum
expert estimate is less than the 95[th] percentile of *P(M)* and for all formations, the
highest value of *P(M)* exceeds the maximum expert estimate by between 40 % and

191     120%.



The 5th to 95th percentiles expressed as a percentage of the median value of *P(M)*
can range from 8-62% for the 5th percentile and 170-307% for the 95th percentile
(the expert estimates show a similar range; Table 1). Figure 3 shows the range of
*P(M)* against the number of unique values for the surface area, thickness, net:gross
and porosity. Surface area and thickness coincide because there are the same
number of unique values for all formations.

**4. Discussion**

The storage capacity estimates of 7 saline aquifers by a group of experts shows that
any single estimate by 1 expert might be a gross under or overestimation of the
median storage capacity. Even using a cohort of experts to provide independent
estimates of the storage capacity does not cover the full range of possible values
using just the data that those same experts collected. In particular, the range of
expert estimates underestimated the highest values of the storage capacity by at
least 40% (and up to 120%). As there is no reasons to assume that any one
combination of variables is more or less likely than any other, all possible
combinations must be assumed to have the same probability. Hence the storage
capacity calculated using all minimum or maximum values for all variables are
equally likely as any other individual combination, though there are more
combinations of variables that will produce storage capacities around the median
value than the extremes, making an estimate around the median more likely overall.

The number of experts in the study was necessarily limited, however using more
experts would not alter the outcome of the study. More experts may increase the
range of estimates produced, but would certainly not decrease it. Having more
experts might be predicted to decrease the standard deviation of the mean
estimate, however, as above, there is no reason to consider that the mean estimate
is a better estimate of the true (unknown) value of the storage capacity than any
other value.

It is therefore evident that the uncertainty associated with a single estimate of $CO_2$
storage capacity for a saline aquifer is large compared to the precision with which at
least some published values are presented. Given both the small database upon
which estimates are typically based, and the inherent variability of the geological
parameters involved, the result is perhaps not surprising, a result confirmed by more
statistically rigorous studies e.g. Keating et al. (2011). The exercise upon which the
present paper is based was conducted using only publicly available data. The experts
had access to a science library, and to the internet. It is apparent that the vast
majority of the data were derived by web-searching, including in most cases the data
from the library which must obviously be located before it can be consulted. A
source of uncertainty within the estimates is therefore the choice of search terms
entered into internet search tools, which could be crucial in either locating or
missing key data sources. In this study, porosity tends to have fewer independent
sources in the literature than the other parameters, leading to potential
underestimation of the uncertainty in comparison to other parameters and hence a
smaller range of calculated storage capacity values for this parameter. The ability to
calculate the uncertainty in a storage capacity estimate is therefore limited by data
availability and uncertainty is likely to be underestimated if this is not taken into
account. In the case of the Mains Formation, the range of calculated capacities is
comparable to the median value (Fig. 3), as all the experts located a single published
porosity value. In other words, the range of storage estimates is partly controlled by
the number of published values, and their accessibility or ease of location. In an
extreme case as with the Mains Formation, the range of *P(M)* is likely to be
underestimated.

A further potential source of variability in the storage estimates is the influence of
the individual assessors.  Both personal judgement and previous experience have
been shown to influence geological interpretation (Polson and Curtis, 2010). In this
case, personal judgement is exercised when faced with parameters for which several
data values are available, with no indication of which are more representative of the
regional mean, and with no objective method of ranking the precision or importance
of the values. One approach under these circumstances is simply to average the
available values; the resulting mean clearly depends on which data have been
located by the individual expert.

Personal judgement is required when estimating net:gross ratio, as the most
common source of data are borehole logs with a summary lithology column showing
whether the sediments within the reservoir interval are interpreted as sandstone,
silty sandstone, siltstone or mudstone (there are no significant limestones in the
study area). Clearly mudstone is non-reservoir, and sandstone is potentially
reservoir, but a more-or-less arbitrary boundary between the two must be drawn. A
more experienced wireline log interpreter might choose to ignore the summary
lithology column of the composite log, and choose a value of, for example, the
gamma ray log as an arbitrary cut-off between reservoir and non-reservoir, or
estimate porosity (see below) and use an arbitrary minimum value of c. 10 % for
reservoir.

The most important control on the quality of the estimate of reservoir thickness is
probably the number of borehole logs used to estimate the mean value. The most
commonly used sources of data in this study (Cameron, 1993; Richards et al, 1993),
typically present 3 summary borehole logs of each storage unit. However the experts
had access to 28 other composite (summary) borehole logs from the region, released
by the UK Government. Some experts choose to use the entire suite of logs
provided, others used only a subset. Even if all logs are used, it is possible to use a
range of methods to calculate mean regional thickness. For example, one can simply
calculate the mean of the storage unit thickness data; or one could to construct a
map and interpolate contours, then estimate mean thickness by some simple
graphical method involving dividing the storage unit into zones of constant thickness
interval and calculating an average thickness weighted to the areas of the zones.  It is
also possible to use commercial software to perform both the contouring and the
reservoir volume calculation, in which case calculating the mean thickness is
unnecessary. Each of these approaches will result in different estimates of the
thickness of the reservoir (or final gross reservoir volume).

For porosity, literature values can be utilised if they exist, though if a range is given
then the mean must be estimated. Sometimes porosity data are only provided
graphically (as a cross-plot of porosity versus log permeability) and the mean value
can only be estimated visually as the points are frequently too dense to be read
individually from the graphs. Alternatively porosity can be calculated from borehole
logs using standard methods - using Formation Density Compensated (FDC) and
Compensated Neutron (CNL) logs for example - either manually or by using
petrophysical computer software if the wireline logs are available in digital form.
Again, the choice of method will influence the result. Measured porosity data are
most commonly from within hydrocarbon fields, where the spatial density of
boreholes is greatest. Whether the porosity of oilfield reservoirs is representative of
the associated aquifer, or is systematically higher and thus introduces a systematic
error in the estimate of aquifer porosity, is a controversial issue (e.g. Wilkinson and
Haszeldine, 2011) for which a judgement is necessary. In a commercial study, it is
possible to purchase porosity data measured from borehole core; unsurprisingly
none of the experts chose this option in this study.

The study reported here could be considered to be typical of regional studies
conducted with the aim of ascertaining which geological units in a region are worthy
of further study, i.e. a scoping study. The data available to the experts will be only a
fraction of the total data collected from the area, and the data must obviously be
located before being utilised. In any hydrocarbon province, it is unlikely that all
possible data can be used in a regional scoping study, due to the large (often very
large) volumes of data that have been collected historically, and due to the non-
availability of some (or much) of the data due to commercial confidentiality. Unless
there are previously published syntheses of data with calculated averages of
parameters such as the thickness of storage units, then some proportion of the total
data will be selected and utilised, inherently introducing uncertainty into the result.
Furthermore, the experts in this study could not spend unlimited periods of time
searching for data, or in processing it once obtained. Again, this restriction is likely to
be encountered in a regional scoping study, where many potential stores must be
assessed within a fixed budget. The North Sea is also typical of hydrocarbon
provinces in that there are a large number of boreholes drilled into relatively small
areas (i.e. producing hydrocarbon fields) and relatively small numbers of boreholes
in the much larger intervening areas. The spacing of the boreholes (data density) is
probably not atypical of other offshore hydrocarbon provinces, though onshore
hydrocarbon provinces may have much higher borehole densities (i.e. boreholes per
square kilometre). Borehole records in the UK are released by the Government, so
that the density of available data may be comparable to other areas of the world
where borehole density is greater but where drilling results are not so readily
available due to commercial confidentiality.

While the uncertainty of estimated storage capacities will vary from study to study,
and can be reduced by costly data collection (or possibly purchase) for any given
geological unit, the results here suggests that there is significant uncertainty in any
storage capacity estimate that does not include a site-specific estimate of
uncertainty.  Note that this analysis does not take account of uncertainty in $CO_2$
density or storage efficiency. Storage efficiency, unless constrained on a unit-by-unit
basis, can introduce an order-of-magnitude uncertainty to a storage estimate (e.g.
Scottish Centre for Carbon Storage, 2009). The geological variability of a storage unit
hence appears to impart less uncertainty into the storage estimate than the storage
efficiency.

It is not possible to estimate the likely uncertainty of any single storage capacity
estimate as there is no way to know whether it is at lower, middle of upper range of
*P(M)*. However, these results show that the storage capacity could range from less
than 10% to over 300% of any single value. This is considerably larger than
uncertainty imparted by the inherent variability within a single, well-constrained
data set, where a study by Deng et al. (2012) found only a 4% uncertainty at 95%
confidence. However, the same study found that incorporating uncertainty in the
capacity estimate reduced the overall storage capacity by over 60% compared to an
earlier study using single values of input parameters. This supports the
recommendation of Chadwick et al. (2008) that a (single) calculated storage capacity
that is similar to the quantity of $CO_2$ to be stored should be regarded as a cautionary
indicator for the suitability of a storage unit for a particular project.

Data for this study were limited to that in the public domain which is probably
realistic for a regional study, where a potentially large number of candidate aquifers
are assessed for first-order suitability for storage (e.g. Scottish Centre for Carbon
Storage, 2009). It is probably not applicable to a detailed study of a single aquifer,
where every effort is made to reduce key uncertainties and where confidential data
may be available. For example, in the estimation of aquifer thickness, every borehole
log that penetrates the storage unit could be utilised, removing the subjective
element of choice associated with taking a subset of the available data. It is also
likely that a more rigorous approach to uncertainty would be used in a single aquifer
study, generating a reliable estimate of the likely range of capacity (e.g. Keating et
al., 2011; Pawar et al., 2016). For this reason, the range of uncertainty for a detailed,
single aquifer study should be substantially less than that derived here, and more
comparable to the 4% relative uncertainty at the 95 % confidence interval found in a
detailed study by Deng et al. (2012).

**5. Conclusions**

The average standard deviation in $CO_2$ capacity for the storage units studied here is ±
64 %. This is substantially greater than the implied precision of many published
storage estimates. The geological uncertainty of a single storage capacity estimate
for a storage unit with no other assessment of uncertainty might be in the range of
30 – 245 % of the estimated value, or 6 to 520 % more conservatively . For storage
units where capacity is on the borderline of being economic or otherwise useable,
this uncertainty may materially influence the decision of acceptance or rejection of
the candidate unit. It should also be recognised that the analysis here does not
exclude the possibility of the useable, real-world, storage capacity of a candidate
storage unit being zero, due to for example, an unfixable leakage pathway or
regulatory issues.

Uncertainty documented in this study is due to a mixture of spatial variability in the
parameters combined with only limited availability of data; the number of
independent (prior) estimates that are located for each parameter; and the variation
in interpretation of the same data by different experts. The range and standard
deviation values in this study should be considered to be minimum values. The
overall uncertainty is likely to be significantly larger as several sources of uncertainty
are not accounted for in this study, in particular uncertainty due to storage efficiency
could be larger than the geological uncertainty assessed here. Therefore a single
assessment of a storage capacity of a geological unit, with no associated assessment
of uncertainty, should be considered to have at least this degree of uncertainty in
the absence of other information.

**Author contribution**

MW designed the initial concept and supervised the storage assessment exercise. DP
performed the majority of the data analysis and interpretation.

**Competing interests**

The authors declare that they have no conflict of interest.

**Acknowledgements**

Thank you to all of the students of the Carbon Capture and Storage Masters of
Science Degree (2009 – 2010) at the University of Edinburgh who gave permission
for their results to be used in this paper. Borehole logs were from the Common Data
Access database, which was kindly made available by Schlumberger.

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

Table 1 – Range of individual expert and distribution (*P(M)*) of storage capacity
estimates. Numbers if brackets are values express as a percentage of the median.

| Storage unit | Expert Median (Mt $CO_2$) | Expert Min (Mt $CO_2$) | Expert Max (Mt $CO_2$) | P(M) Median (Mt $CO_2$) | P(M) 5th percentile (Mt $CO_2$) | P(M) 95th percentile (Mt $CO_2$) |
|---|---|---|---|---|---|---|
| Burns Sandstone Member | 1905 | 119 (6%) | 5381 (282%) | 1755 | 144 (8%) | 5035 (287%) |
| Beatrice Formation | 120 | 37 (31%) | 192 (160%) | 110 | 25 (23%) | 202 (185%) |
| Orrin Formation | 96 | 18 (18%) | 785* (819%) | 102 | 16 (16%) | 179 (176%) |
| Mains Formation | 197 | 95 (48%) | 245 (124%) | 186 | 116 (62%) | 316 (170%) |
| Hopeman Sandstone Formation | 263 | 114 (43%) | 457 (174%) | 220 | 66 (30%) | 490 (223%) |
| Findhorn Formation | 1381 | 565 (40%) | 3632 (263%) | 1471 | 626 (43%) | 3431 (233%) |
| Strath Rory Formation | 763 | 75 (10%) | 2300 (302%) | 724 | 75 (10%) | 2219 (307%) |

* This is significantly higher than the 95th percentile due to 1 expert estimating the
volume of the formation to be significantly higher than the other experts.

**Figure Legends**

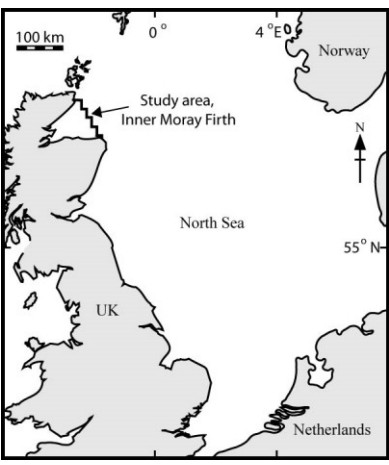


Figure 1 – location map of study area.

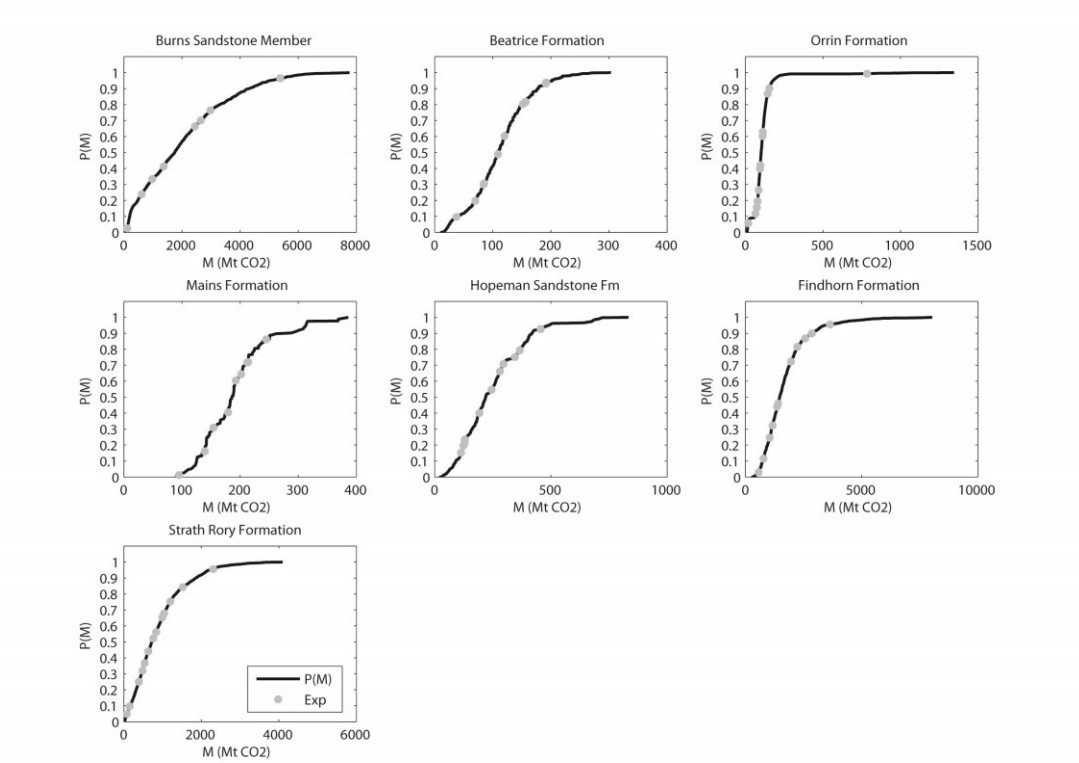


Figure 2. Range of storage capacity estimates using the different values for variables
found by group of experts for 7 saline aquifers. Range is shown as a cumulative
density function but does not represent the true probability density function for
each aquifer.

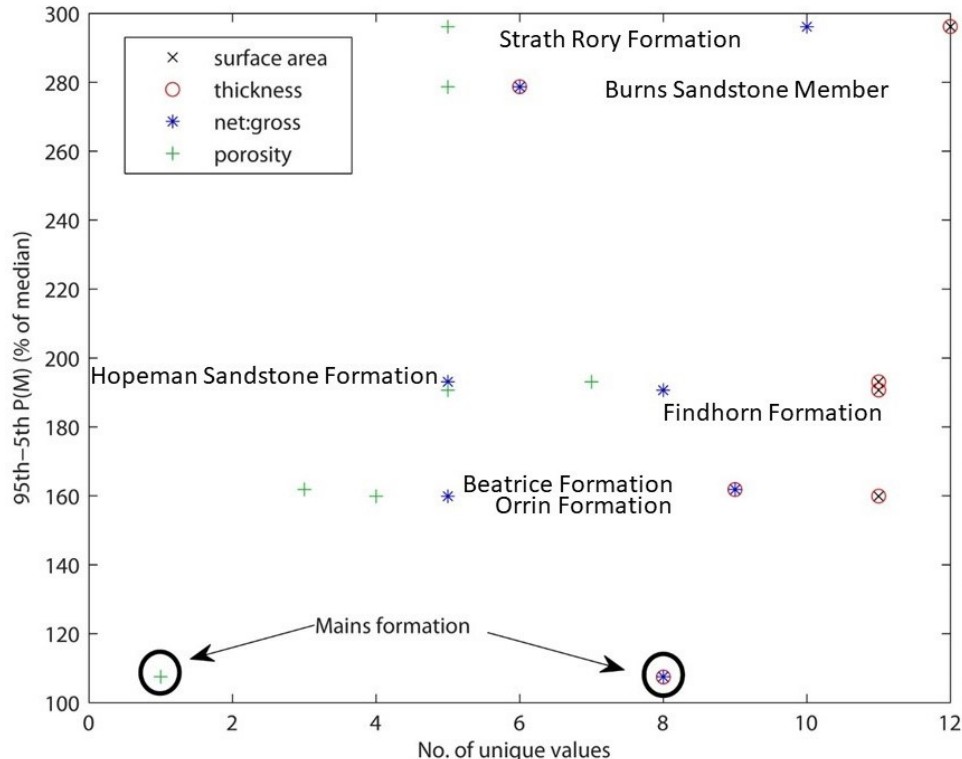


Figure 3. The Range of *P(M)* (5th -95th percentile) against number of unique values for
the area, thickness, net:gross and porosity.