# Peer review of "Uncertainty in regional estimates of capacity for carbon capture and storage"

_Solid Earth, 2019_

## Referee Comment (RC1) · Ran Calvo (Referee) · 7 Apr 2019

I have read this manuscript with a lot of interest. The presented work is brilliant in its simplicity. Take few students, give them a questionnaire of 3-4 questions (area and thickness or total volume, N:G, and porosity). Calculate total capacity, calculate some statistical values, summarize it and you got a nice paper. My main concern is regarding to the amount of questionnaires in this study. To my humble opinion seven students are not enough to get the full range and not enough to calculate distribution of parameters. Apart for that, excellent work.

Specific remarks: Line 41: Wilkinson et al., 2013. In the bibliographic list I can see only 2010. Line 47: Calvo et al., 2019 is missing in your bibliographic list. Line 54: Medina

et al., 2011 is missing in your bibliographic list. Line 102: I think it will be nice to know how many students are in this study. From your supplementary material I understood that only 7 experts are in this study. In that matter, dose 7 experts are sufficient to conduct such statistically examination? Line 108: You have an extra footnote (1). Line 110: replace "ratio), using the product" into "ratio). The product". Line 120: What are the references #14 and #15? Line 153: Cameron et al., 1990 is missing in your bibliographic list. Line 159: Usually Formation (capital F) and nut just formation. Line 269: Change & into "and". Lien 320: (e.g. 1). What it that? Figure 3: You can add the names of all aquifers.

---

## Referee Comment (RC2) · Anonymous Referee #2 · 4 Jun 2019

This paper deals with the uncertainty in estimation of storage capacity for CO2 in deep saline aquifers. As a reviewer, I have a few concerns about this paper. The Editor may wish to consider these comments in deciding whether to accept the paper for publication.

Major comments:

(1) When I review papers, by far my most frequent criticism is "the novel contribution of this manuscript is not clear." That criticism pertains here. It is clear that the authors are concerned with uncertainty in estimating CO2 storage capacity, and in coming up with some assessment over whether the uncertainty is large or small. Nevertheless, I am not able to determine the specific knowledge gap or research question being addressed here. The Introduction to the paper does not contain a hypothesis to be tested,

nor a clear statement of a central objective, nor a clear statement of a knowledge gap to be addressed. I do note that the authors state that "an estimate of the accuracy of single-value storage capacities is of practical use," and also that "For this study, an assessment of the accuracy of storage capacity estimates was conducted as part of a study of an area of the UK territorial waters." However, I dispute that the authors have assessed the "accuracy" of any estimates – that would require comparing an estimate to a known or trusted value, which is not done in this paper. Hence, in the end, I find it unclear what important contribution is represented by this manuscript that would warrant its publication.

(2) Closely related to the comment immediately above, it is not clear what we can conclude or take away from the exercise performed here by the authors. Towards the end of the discussion, the authors state that their analysis "is probably realistic for a regional study, where a potentially large number of candidate aquifers are assessed for first-order suitability for storage," but that "it is probably not applicable to a detailed study of a single aquifer, where every effort is made to reduce key uncertainties and where confidential data may be available." I think the implication here is that the estimate of uncertainty made herein would apply to "rough" estimates of storage capacity. Thus the main conclusion seems to be that initial or "rough" estimates of storage capacity carry a high degree of uncertainty. This is not surprising; it is to be expected. Hence, once again, I find it unclear what important new insight is represented by this paper. It is possible that there is an important insight or contribution here, and that I am missing it – that is possible – but if that is the case, the authors must do a better job of clarifying the importance of their work and what it offers to the community.

(3) I question part of the discussion given by the authors. In this analysis, the estimate of storage capacity is made by multiplying together six factors, four of which (A, h, NG, phi) must be estimated independently. The authors correctly note that the maximum possible estimate of storage capacity would be made by multiplying together the highest estimated values of A, h, NG, and phi. The authors also note that none of their

team of experts ever made such an estimate, i.e., in no case did one expert ever make the highest estimates of all four parameters simultaneously. The authors claim that "real" uncertainty may therefore be even greater than the range spanned by their team of experts, because "all possible combinations must be assumed to have the same probability," and "hence the storage capacity estimated using all minimum or maximum values for all variables are equally likely as any other individual combination." I do not think this is correct. First, I think it is unclear that the probability distribution of each parameter individually should be considered to be a uniform pdf. If twelve experts each make a prediction of a variable, and that variable actually has a "true" or "correct" value, is it the case that each of the 12 estimates is equally "good" as the others? Or might we expect that the values closer to the middle of the range are more likely to be "good" estimates than the highest and lowest estimates? If the 12 experts are, in fact, experts, then I think the mean/median values should be "better" estimates (i.e., lie closer to the true or correct value) than the highest and lowest estimates. This would mean that the product of the four highest parameter estimates is *not* just as likely as other combinations. And furthermore, suppose I am wrong on this point – suppose that the probability distributions of the individual parameters can be considered to be uniform pdfs, such that any value between min_value and max_value is equally likely. What does this mean about the probability distribution of the *product* of the values? Is the product of uniform-distributed variables also uniform-distributed? I would guess not! If I roll a standard die, the chances of getting 1, 2, 3, 4, 5, or 6 pips is equally likely. But suppose I roll two dice, and I multiply the values together. The chances of getting a product of 1 or 36 is *not* the same as getting a product of 12. And if I roll four dice, the probability of the product being a 1 is definitely lower than the probability of getting, say, 144 for a product. So for two reasons, I think the authors' contention is mistaken. They may want to consult with an expert in probability or statistics. I would bet that the product of uniform-distributed variables approaches a log-normal distribution or maybe a gamma distribution as the number of variables gets large. (This is just a hunch, I do not know if it is correct.)

(4) It is an interesting question whether 12 experts is enough to represent the range of uncertainty of the individual parameters. Suppose we want to estimate the parameter h and we want to have some quantification of the uncertainty of the estimate. How many expert estimates of h must we obtain before we can conclude that the standard deviation of the estimates is a meaningful quantification of uncertainty? I think 12 sounds like it might be barely enough, but I do not know enough statistics to know the answer to this question – I am just going on intuition. Again the authors may want to consult with an expert in probability or statistics.

Minor comments:

(5) The text on lines 88-98 of the Introduction probably does not belong in the Introduction. Most of this text is either "site description" or "methods". I do not think any of it establishes the main idea of the paper, and therefore I would not put it in the Introduction.

(6) As far as I can tell, the paper does not indicate how many experts were used in this analysis. In my comments above, I assumed 12 experts, but I do not know if this is correct. I am assuming 12 based on the data that I see in Figures 2 and 3. It looks like maybe there were 12 experts involved. But I do not know if this is correct. The actual value should be indicated clearly in the text and possibly in the figure captions too.

Technical corrections:

(7) The grammar needs a little clean-up. The first sentence of the abstract is a run-on sentence because of the peculiarities of the conjunction "however". Later in the abstract the authors state "due a combination of using different published values", i.e., missing the word "to" after "due". Later in the abstract there is another run-on sentence where a comma should be a semi-colon. That is just the abstract. Other grammar issues are found throughout the paper. Maybe hire somebody to read the paper thoroughly and clean up any such errors. They do not impede comprehension, but they distract.

(8) A couple cited papers are not in the reference list. I put this under "minor comments", but it MUST be corrected. Examples are the citations of Calvo et al. (2019) and Medina et al. (2011).

(9) Equation 1 uses h, but the following text uses H.

(10) The word "data" is plural but is incorrectly used as singular throughout the paper.

---

## Author Comment (AC1) · 13 Jun 2019

Thanks for the comments.

Your comments contain 1 science criticism, and several smaller errors in the text. Here is a response to the science, thanks for the text errors, we will correct them.

Q1: "My main concern is regarding to the amount of questionnaires in this study. To my humble opinion seven students are not enough to get the full range and not enough to calculate distribution of parameters."

There were actually 13 questionnaires (we'll make this clear when revising the MS), though not all respondents answered for each storage formation. I agree that more is better in this context, as in most other cases with data. However, considerable

work was involved for each respondent (tens of hours) so getting more volunteers will be tricky. The point of the paper is that the experts came up with a wide range of answers - having more experts might increase the range a little, but will surely not decrease it. Equally, if you do the calculation only once (and a consultancy company for example, asked this question, would surely only do the calculation once), you will not know where is the range of answers you lie. Having more respondents will not change this conclusion.

---

## Author Comment (AC2) · 13 Jun 2019

Thanks for your comments. You raised several points:

1) "The Introduction to the paper does not contain a hypothesis to be tested, nor a clear statement of a central objective, nor a clear statement of a knowledge gap to be addressed." The paper tests the hypothesis that regional storage estimates are subject to considerable uncertainty, despite many published estimates having no indication of this. We'll make this clearer in the revised MS

"I dispute that the authors have assessed the "accuracy" of any estimates – that would require comparing an estimate to a known or trusted value". True – we have assessed the precision. We do not know the true value (it is unknowable) so accuracy cannot be

assessed.

2) "it is not clear what we can conclude or take away from the exercise performed here by the authors." In the same paragraph, the reviewer's says that it is 'obvious' that regional estimates have substantial uncertainty. This may be true, but you would not know that from the published literature, most estimates are single numbers with no indication of uncertainty. As the 'true' value of capacity is unknowable, another approach has to be used to assess reliability – which is what we present. Actually, we can probably make this clearer – if your estimate of storage capacity is similar to your likely $CO_2$ volume to be stored, then there is a substantial probability that you will find that the $CO_2$ will not fit in the store (because you have over-estimated the capacity)!

3) "I think it is unclear that the probability distribution of each parameter individually should be considered to be a uniform pdf." Assessing the pdf of an unknown quantity is always going to be difficult. If 2 experts find the same literature value for a variable, does that make that value more probably than if only 1 expert finds it? Surely not. Conversely, you could argue that if there are 10 different values published for a single parameter (e.g. mean porosity for a formation from 10 different locations) that the best estimate of the regional mean value is the mean of the individual data, making the centre of the distribution more probable than the margins, as with e.g. a normal distribution. For most of the data used in the exercise, there is so little published information that it is not possible to realistically assess the distribution, so a uniform pdf is no worse than any other.

Re: The reviewer's section on rolling dice. We clearly stated this: "though there are more combinations of variables that will produce storage capacities around the median value than the extremes, making an estimate around the median more likely overall." Lines 191 -193.

Re: "They may want to consult with an expert in probability or statistics". One of us (DP) is a professional statistician.

[Figure]

4) "It is an interesting question whether 12 experts is enough to represent the range of uncertainty of the individual parameters." As with the other review, there were actually 13 experts (we'll make this clear when revising the MS), though not all respondents answered for each storage formation. I agree that more is better in this context, as in most other cases with data. However, considerable work was involved for each respondent (tens of hours) so getting more volunteers will be tricky. The point of the paper is that the experts came up with a wide range of answers - having more experts might increase the range a little, but will surely not decrease it. Equally, if you do the calculation only once (and a consultancy company for example, asked this question, would surely only do the calculation once), you will not know where is the range of possible answers you lie. Having more respondents will not change this conclusion.

The other comments do not require replies, thanks for spotting the errors, we will amend suitably.

---

## Author Response (AR1)

Reviewer 1:

Comment: "To my humble opinion seven students are not enough to get the full range and not enough to calculate distribution of parameters."

Reply: There were actually 13 questionnaires, though not all respondents answered for each storage formation. I agree that more is better in this context, as in most other cases with data. However, considerable work was involved for each respondent (tens of hours) so getting more volunteers will be tricky. The point of the paper is that the experts came up with a wide range of answers - having more experts might increase the range a little, but will surely not decrease it. Equally, if you do the calculation only once (and a consultancy company for example, asked this question, would surely only do the calculation once), you will not know where is the range of answers you lie. Having more respondents will not change this conclusion.

Changes: added to para 1 of discussion: "The number of experts in the study was necessarily limited, however using more experts would not alter the outcome of the study. More experts may increase the range of estimates produced, but would certainly not decrease it. Having more experts might be predicted to decrease the standard deviation of the mean estimate, however, as above, there is no reason to consider that the mean estimate is a better estimate of the true (unknown) value of the storage capacity than any other value."

Specific remarks:

Line 41: Wilkinson et al., 2013. In the bibliographic list I can see only 2010.  **REF added**

Line 47: Calvo et al., 2019 is missing in your bibliographic list. **Ref added**

Line 54: Medina et al., 2011 is missing in your bibliographic list. **Ref added**

Line 102: I think it will be nice to know how many students are in this study. From your supplementary material I understood that only 7 experts are in this study. In that matter, dose 7 experts are sufficient to conduct such statistically examination? **See above, the number of experts has been added to the abstract, methods and other sections.**

Line 108: You have an extra footnote (1).  **deleted**

Line 110: replace "ratio), using the product" into "ratio). The product". **Done**

Line 120: What are the references #14 and #15? **Fixed**

Line 153: Cameron et al., 1990 is missing in your bibliographic list. **Should be 1993, fixed**

Line 159: Usually Formation (capital F) and nut just formation. **Fixed**

Line 269: Change & into "and". **Fixed (and 11 others)**

Lien 320: (e.g. 1). What it that? Reference is **SCCS(2009), changed**

Figure 3: You can add the names of all aquifers. **Done**

**Reviewer 2** raised several points:

1) Comment: "The Introduction to the paper does not contain a hypothesis to be tested, nor a clear statement of a central objective, nor a clear statement of a knowledge gap to be addressed."

Reply: The paper tests the hypothesis that regional storage estimates are subject to considerable uncertainty, despite many published estimates having no indication of this. We'll make this clearer in the revised MS.

Changes: Abstract "Here, we test the hypothesis that the uncertainty in such estimates is a significant proportion of the estimated storage capacity, and should hence be evaluated as a part of any assessment." Similar text has been added to the introduction.

Comment: "I dispute that the authors have assessed the "accuracy" of any estimates – that would require comparing an estimate to a known or trusted value".

Reply: True – we have assessed the precision. We do not know the true value (it is unknowable) so accuracy cannot be assessed.

Changes: We've replaced 'accuracy' with either uncertainty or precision in the text, depending on the context.

2) Comment: "it is not clear what we can conclude or take away from the exercise performed here by the authors."

Reply: In the same paragraph, the reviewer's says that it is 'obvious' that regional estimates have substantial uncertainty. This may be true, but you would not know that from the published literature, most estimates are single numbers with no indication of uncertainty. As the 'true' value of capacity is unknowable, another approach has to be used to assess reliability – which is what we present. Actually, we clearly state the implications – if your estimate of storage capacity is similar to your likely $CO_2$ volume to be stored, then there is a substantial probability that you will find that the $CO_2$ will not fit in the store (because you have over-estimated the capacity)!

Changes: none, we think we had this covered.

3) Comment: "I think it is unclear that the probability distribution of each parameter individually should be considered to be a uniform pdf."

Reply: Assessing the pdf of an unknown quantity is always going to be difficult. If 2 experts find the same literature value for a variable, does that make that value more probably than if only 1 expert finds it? Surely not. Conversely, you could argue that if there are 10 different values published for a single parameter (e.g. mean porosity for a formation from 10 different locations) that the best estimate of the regional mean value is the mean of the individual data, making the centre of the distribution more probable than the margins, as with e.g. a normal distribution. For most of the data used in the exercise, there is so little published information that it is not possible to realistically assess the distribution, so a uniform pdf is no worse than any other.

Reply: no changes, again, we think this was covered sufficiently

Re: comment: The reviewer's section on rolling dice.

Reply: We clearly stated this: "though there are more combinations of variables that will produce storage capacities around the median value than the extremes, making an estimate around the median more likely overall." Lines 191 -193.

Changes: none.

Re: "They may want to consult with an expert in probability or statistics".

Reply: We're confident this is OK.

Changes: none

Comment 4) "It is an interesting question whether 12 experts is enough to represent the range of uncertainty of the individual parameters."

Reply; see the same question for reviewer 1, above.

**Smaller points:**

**Comment**: The first sentence of the abstract is a run-on sentence because of the peculiarities of the conjunction "however".

Reply: debatable, I've Googled what a 'run-on sentence' is, and this does not appear to be one. It's a bit long though, so split at 'however':

Change: "Carbon capture and storage (CCS) is a potentially important technology for the mitigation of industrial $CO_2$ emissions. However the majority of the subsurface storage capacity is in geological strata for which there is relatively little information, the so-called saline aquifers."

**Comment**: Later in the abstract the authors state "due a combination of using different published values", i.e.,missing the word "to" after "due".
Reply: typo, **fixed**

**Comment**: Later in the abstract there is another run-on sentence where a comma should be a semi-colon.
Reply: again, too long but not technically a run-on?
Change: split the sentence: "The range of storage estimates produced by the experts shows that there is significant uncertainty in such estimates, in particular the experts' range does not capture the highest possible capacity estimates. This means that by not accounting for uncertainty, such regional estimates may underestimate the true storage capacity."

Comment: Other grammar issues are found throughout the paper. Maybe hire somebody to read the paper thoroughly and clean up any such errors. They do not impede comprehension, but they distract.
Reply: we've read it through and are happy with it.

Comment 8) A couple cited papers are not in the reference list. I put this under "minor comments" but it MUST be corrected. Examples are the citations of Calvo et al. (2019) and Medina et al. (2011). **Sorted, as above**

(9) Equation 1 uses h, but the following text uses H. **Sorted (to 'h')**

(10) The word "data" is plural but is incorrectly used as singular throughout the paper.

These are examples I can find of where singular / plural is implied in the uses of the word 'data':

"data **are** usually available" – plural already

"Subsurface geological data **were**" – plural already

"or missing key data **sources**" – plural already

"for which several data values **are** available" – plural already

"on which data **have** been" – plural already

"The most commonly used **sources** of data" – plural already

 "Sometimes porosity data **are**" – plural already

"Measured porosity data **are**" – plural already

And so on. I can only find the following example of the problem:

"Data for this study **was** limited", changed to "Data for this study **were** limited" (line 330 of revised MS).

[revised manuscript text omitted]

---

## Referee Report (RR1)

se-2019-45

Title: "**Uncertainty in regional estimates of capacity for carbon capture and storage"**

This paper describes the uncertainty in storage capacity estimates found by a group of experts for a set of CO2 storage reservoirs (saline aquifers) in the North Sea. The experts are graduate students who were given access to a library and the internet. Limited data for many of the reservoirs leads to wide estimates of storage capacity. The work is interesting, and highlights the need for site specific data and probabilistic methods to make better estimates. I am suggesting that references be added for completeness.

Specific Comments:

1) Line 38. Remove the 'so-called". Although I believe that these should be called brine reservoirs (not potable water), the ship has left the dock on this one, and saline aquifer is used extensively in the literature and has redefined the original definition of an aquifer as being potable water.
2) Somewhere in lines 41-51 you should discuss the NatCarb database that was created by the US Department of Energy for the entire US.
   **https://www.netl.doe.gov/coal/carbon-storage/strategic-program-support/natcarb-atlas**
3) Somewhere in the introduction you should reference some of the early work on storage capacity by Stefan Bachu, for example:

   Bachu, S. Sequestration ofCO2 in geological media: criteria and approach for site selection in response to climate change. Energy Convers. Manage. 2000, 41, 953–970.

4) Line 59, another 'so-called'
5) Consider adding references to the following papers that describe probabilistic approaches to estimating storage capacity for multiple regional sinks, perhaps even a short paragraph on the topic:

   2011 Keating, G, R.S. Middleton, P.H. Stauffer, H.S. Viswanathan, B.C. Letellier, P Pasqualini, R. Pawar, A.W. Wolfsberg, Meso-scale carbon sequestration site screening and CCS infrastructure analysis, *Environ. Sci. Technol.*, (JAN 1 2011) Vol.45, iss.1, p.215-222

   2012 Middleton, R.S., G. Keating, P.H. Stauffer, A. Jordan, H. Viswanathan, Q. Kang, B. Carey, M. Mulkey, J. Sullivan, S.P. Chu, and R. Esposito, The multiscale science of CO2 capture and storage: From the pore scale to the regional scale. Energy and Environmental Science, 5,7328 | doi:10.1039/C2EE03227A.

   2012 Middleton, R.S., Keating, G.N., Stauffer, P.H., Viswanathan, H.S., Pawar, R.J., Effects of geologic reservoir uncertainty on CO2 transport and storage infrastructure. Int. J. Greenhouse Gas Control, doi:10.1016/j.ijggc.2012.02.005.

2016 Pawar, R.J., G. Bromhal, S.P. Chu, R.M. Dilmore, C. Oldenburg, P.H. Stauffer, Y. Zhang, G. Guthrie, The National Risk Assessment Partnership's Integrated Assessment Model for Carbon Storage:  A Tool to Support Decision Making Amidst Uncertainty,  Int. J. Greenhouse Gas Control, 52, 175–189.

6) Line 82. Instead of saying 'too small' perhaps it would be more precise to say "may be rejected as having too low a storage capacity . . . "
7) Line 339  Consider referencing studies that show how regional estimates often over-estimate storage capacity when refined to include site specific data

2012 Deng, H., P.H. Stauffer, Z. Dai, Zunsheng Jaio, R.S. Surdam, Simulation of Industrial-Scale $CO_2$ Storage: Multi-Scale Heterogeneity and its Impacts on Storage Capacity, Injectivity and Leakage, Int. J. Greenhouse Gas Control, Volume 10, September 2012, Pages 397–418.

2014 Dai, Z., P. H. Stauffer, J.W. Carey, R.S. Middleton, Z. Lu, J.F. Jacobs, K. Hnottavange-Telleen, L.H. Spangler, Pre-site characterization risk assessment for commercial-scale carbon sequestration, Environ. Sci. Technol, DOI: 10.1021/es405468.

2019    Onishi, T., M.C. Nguyen, J.W. Carey, B. Will, W. Zaluski, D.W. Bowen, B.C. Devault, A. Duguid, Quanlin Zhou, S.H. Fairweather, L.H. Spangler, and P.H. Stauffer, Potential $CO_2$ and brine leakage through wellbore pathways for geological $CO_2$ sequestration using the National Risk Assessment Partnership Tools: Application to the Big Sky Regional Partnership, Int. J. Greenhouse Gas Control, https://doi.org/10.1016/j.ijggc.2018.12. 002

---

## Author Response (AR2)

**Wilkinson and Polson, Response to Reviewer 3 and Editors comments.**

Reviewer 3:

1) Line 38. Remove the 'so-called". **Done. There was one in the abstract too, I removed it.**

2) Somewhere in lines 41-51 you should discuss the NatCarb database that was created by the US Department of Energy for the entire US. [https://www.netl.doe.gov/coal/carbon-storage/strategic-programsupport/natcarb-atlas](https://www.netl.doe.gov/coal/carbon-storage/strategic-programsupport/natcarb-atlas) **ADDED lines 43-45**

3) Somewhere in the introduction you should reference some of the early work on storage capacity by Stefan Bachu, for example:
Bachu, S. Sequestration ofCO2 in geological media: criteria and approach for site selection in response to climate change. Energy Convers. Manage. 2000, 41, 953–970. **DONE line 49**

4) Line 59, another 'so-called' **REMOVED**

5) Consider adding references to the following papers that describe probabilistic approaches to estimating storage capacity for multiple regional sinks, perhaps even a short paragraph on the topic:
2011 Keating, G, R.S. Middleton, P.H. Stauffer, H.S. Viswanathan, B.C. Letellier, P Pasqualini, R. Pawar, A.W. Wolfsberg, Meso-scale carbon sequestration site screening and CCS infrastructure analysis, Environ. Sci. Technol., (JAN 1 2011) Vol.45, iss.1, p.215-222 **Great reference, added line 51**

2012A Middleton, R.S., G. Keating, P.H. Stauffer, A. Jordan, H. Viswanathan, Q. Kang, B. Carey, M. Mulkey, J. Sullivan, S.P. Chu, and R. Esposito, The *cross*-scale science of CO2 capture and storage: From the pore scale to the regional scale. Energy and Environmental Science, 5,7328 | doi:10.1039/C2EE03227A. **Also added. Line 51**

2012B Middleton, R.S., Keating, G.N., Stauffer, P.H., Viswanathan, H.S., Pawar, R.J., Effects of geologic reservoir uncertainty on CO2 transport and storage infrastructure. Int. J. Greenhouse Gas Control, doi:10.1016/j.ijggc.2012.02.005. **ADDED**

2016 Pawar, R.J., G. Bromhal, S.P. Chu, R.M. Dilmore, C. Oldenburg, P.H. Stauffer, Y. Zhang, G. Guthrie, The National Risk Assessment Partnership's Integrated Assessment Model for Carbon Storage: A Tool to Support Decision Making Amidst Uncertainty, Int. J. Greenhouse Gas Control, 52, 175–189. **ADDED line 66**

6) Line 82. Instead of saying 'too small' perhaps it would be more precise to say "may be rejected as having too low a storage capacity . . . " **DONE**

7) Consider referencing studies that show how regional estimates often overestimate storage capacity when refined to include site specific data

2012 Deng, H., P.H. Stauffer, Z. Dai, Zunsheng Jaio, R.S. Surdam, Simulation of Industrial-Scale CO2 Storage: Multi-Scale Heterogeneity and its Impacts on Storage Capacity, Injectivity and Leakage, Int. J. Greenhouse Gas Control, Volume 10, September 2012, Pages 397–418. **DONE line 346, 366.**

2014 Dai, Z., P. H. Stauffer, J.W. Carey, R.S. Middleton, Z. Lu, J.F. Jacobs, K. Hnottavange-Telleen, L.H. Spangler, Pre-site characterization risk assessment for commercial-scale carbon sequestration, Environ. Sci. Technol, DOI: 10.1021/es405468. **ADDED line 64**

2019 Onishi, T., M.C. Nguyen, J.W. Carey, B. Will, W. Zaluski, D.W. Bowen, B.C. Devault, A. Duguid, Quanlin Zhou, S.H. Fairweather, L.H. Spangler, and P.H. Stauffer, Potential CO2 and brine leakage through wellbore pathways for geological CO2 sequestration using the National Risk Assessment Partnership Tools: Application to the Big Sky Regional Partnership, Int. J. Greenhouse Gas Control, https://doi.org/10.1016/j.ijggc.2018.12. 00

**This reference is too off-topic, it does not concern storage capacity as far as I can see. I have not included it.**

**Editors comments:**

My only concern (also raised by the reviewers) is that the literature review presented is slightly weak. Please address this issue as well as the comments by Referee #3.

The extra references added from Reviewer 3's efforts should address the lit review side of things.